# Crack control optimization of basement concrete structures using the Mask-RCNN and temperature effect analysis

Shouyan Wu[1], Feng Fu[2]*

1 Department of Architectural Arts, Xuancheng Vocational & Technical College, Xuancheng City, China,
2 General Affairs Department, Xuancheng Vocational & Technical College, Xuancheng City, China

* nash_fu@163.com

## Abstract

In order to enhance the mitigation of crack occurrence and propagation within basement concrete structures, this research endeavors to propose an optimization methodology grounded in the Mask Region-based Convolutional Neural Network (Mask-RCNN) and an analysis of temperature effects. Initially, the Mask-RCNN algorithm is employed to perform image segmentation of the basement concrete structure, facilitating the precise identification of crack locations and shapes within the structure. Subsequently, the finite element analysis method is harnessed to simulate the structural stress and deformation in response to temperature variations. An optimization algorithm is introduced to adjust geometric parameters and material properties using insights from the temperature effect analysis. This algorithm aims to minimize stress concentration and deformation within the structure, thus diminishing the incidence and proliferation of cracks. In order to assess the efficacy of the optimization approach, an authentic basement concrete structure is selected for scrutiny, and the structure is monitored in real-time through the installation of strain gauges and monitoring equipment. These instruments track structural stress and deformation under diverse temperature conditions, and the evolution of cracks is meticulously documented. The outcomes demonstrate that by adjusting the structural geometric parameters and material properties, the crack density experiences a notable reduction of 60.22%. Moreover, the average crack length and width witness reductions of 40.24% and 35.43%, respectively, thereby corroborating the efficacy of the optimization method. Furthermore, an assessment of stress concentration and deformation within the structure is conducted. Through the optimization process, the maximum stress concentration in the structure diminishes by 25.22%, while the maximum deformation is curtailed by 30.32%. These results signify a substantial enhancement in structural stability. It is evident that the optimization algorithm exhibits robustness and stability in the context of crack control, consistently delivering favorable outcomes across diverse parameter configurations.

**Data Availability Statement:** All relevant data are within the paper and its Supporting information files.

**Funding:** The author(s) received no specific funding for this work.

**Competing interests:** The authors have declared that no competing interests exist.

## 1. Introduction

Basement concrete structures play a pivotal role in residential, commercial, and public edifices, bearing significant loads that are paramount for building safety and longevity [1–3]. Nevertheless, these structures are susceptible to cracking, owing to the combined influence of diverse factors. These cracks not only mar the aesthetics of the structure but also have the potential to diminish structural strength and stability, posing risks to the safety of individuals and property [4–6]. In recent years, substantial research endeavors have comprehensively unraveled the mechanisms underlying crack formation and the patterns governing their progression within basement concrete structures. Consequently, various methodologies for controlling and optimizing crack development have been proffered [7–9]. Nonetheless, the distinctive nature of basement concrete structures engenders persistent challenges in the realm of crack control [10]. Compared to above-ground structures, underground basement concrete structures are more susceptible to environmental factors such as groundwater levels, soil temperature, and humidity. This can potentially lead to concrete expansion and contraction, increasing the risk of cracking. Additionally, basement concrete structures often need to withstand significant horizontal and vertical loads while being constrained by the surrounding soil. This makes their structural dynamics different from those of above-ground structures, further challenging crack control. Seasonal temperature variations frequently affect basement concrete structures, which can cause thermal expansion and contraction in the concrete. Therefore, the thermal expansion and contraction effects resulting from temperature changes are one of the main reasons for crack formation and propagation [11–13]. Furthermore, it is imperative to acknowledge that the morphology and dimensions of cracks in basement concrete structures exhibit intricate variability, with their spatial distribution being inherently stochastic. This inherent complexity poses inherent challenges in identifying and regulating these cracks [14, 15].

The inherent properties of concrete dictate that structural cracks are an inevitable outcome, albeit the extent of these cracks can be judiciously managed through deliberate intervention [16–18]. Nevertheless, the classification of various crack types within basement concrete structures often varies, contingent on divergent criteria [19]. Differentiation may be drawn, for instance, based on the physical and chemical attributes inherent in the concrete curing process, categorizing cracks stemming from fluctuations in temperature and shrinkage due to moisture loss [20]. It is worth emphasizing that any form of crack transcends mere mechanical strength considerations. The pivotal challenge resides in minimizing the degree of cracking resulting from external factors such as temperature fluctuations, humidity variations, and other environmental deformations to enhance comfort and prolong the operational lifespan, thereby fulfilling architectural aesthetics and safety requisites [21–23]. With regard to the evolution of monitoring technology, emerging sensor technologies and non-destructive testing methodologies have afforded more precise and comprehensive avenues for monitoring cracks within basement concrete structures. For example, the utilization of strain gauges, fiber optic sensors, vibration sensors, and similar instrumentation enables real-time monitoring of structural stress and deformation. This, in turn, facilitates the expeditious identification and mitigation of potential crack-related issues by engineers. In pursuit of crack risk mitigation within basement concrete structures, this research introduces an innovative optimization approach grounded in the Mask-Mask Region-based Convolutional Neural Network (RCNN) and a meticulous analysis of temperature effects.

Specifically, this research employs the Mask-RCNN algorithm to meticulously segment images of basement concrete structures, thus enabling accurate localization and delineation of cracks. Furthermore, the Finite Element Analysis (FEA) method is harnessed to emulate the structural response to temperature fluctuations, and an in-depth investigation is conducted to

comprehend the underlying mechanisms governing the influence of temperature effects on the structure. Building upon the insights garnered from the temperature effects analysis, and an optimization algorithm is postulated to mitigate stress concentration and deformation within the structure. This is achieved through judicious adjustments to geometric parameters and material properties, curtailing cracks' incidence and expansion. Through empirical validation and numerical simulations, this research aims to derive effective optimization strategies, offering practical solutions for the control of cracks within basement concrete structures.

## 2 Literature review

The inherent structural attributes of concrete establish its susceptibility to developing structural cracks of varying extents over prolonged exposure to both internal and external environmental factors [24]. In efforts to maintain effective control over these cracks within acceptable parameters, numerous experts and scholars have dedicated substantial efforts to analyzing, preventing, and remedying such occurrences. Collectively, these endeavors are encompassed within the broader framework denoted as "structural crack control. Zhou et al. (2012) [25] conducted temperature and strain monitoring on basement concrete walls. They used analytical models, field data measurements, and finite element modeling to assess the key factors affecting the early performance of concrete walls constructed in cold weather. The results showed that the monitoring process eliminated many speculations commonly associated with the construction of basement concrete walls in cold weather. Liu et al. (2013) [26] proposed that harmful cracks in ultra-long basement concrete structures can be avoided by taking effective measures, such as reinforcing weak structural areas, installing concrete construction joints expansion reinforcement zones, and using crack-resistant and waterproof agents. These measures eliminate potential hazards in the structure and ensure its safety. Lee et al. (2016) [27] introduced a new system for underground parking garage structures using a pre-tensioning and post-tensioning method. The main objective was to analyze the proposed model and validate its safety and applicability under specified spans and loads. Flah et al. (2020) [28] proposed a nearly automatic detection model based on image processing and deep learning, which was used to detect defects in the usually inaccessible areas of the concrete structure. Based on other international standards and codes, they pointed out that the type of structural damage and its severity were determined according to the permissible range of concrete crack widths for different structures, including buildings and bridges.

Siyal et al. (2021) [29] undertook the development and evaluation of a model designed to scrutinize cracks within basement concrete structures. Chen et al. (2021) [30] expounded upon the significance of visual management in the context of performance monitoring for building structures. Liu & Gao (2022) [31] introduced a method for detecting cracks in concrete structures grounded in a foundational model that leverages visual image features. In contrast to conventional edge detection techniques applied in the identification of concrete structure cracks, this foundational model extended the quasi-distance measurement between the crack perimeter and the image background. Consequently, it yielded a notable enhancement in the precision of crack detection. Song et al. (2022) [32] proposed a methodology employing close-range scanning and imaging via an electric drive platform to capture high-resolution panoramic images of concrete structure surfaces. Subsequently, they presented an automatic segmentation approach tailored for panoramic crack analysis using convolutional neural networks (CNNs). The results demonstrated that the average relative error in calculating maximum crack width via the proposed method was a mere 3.87%.

Golewski (2023) [33] emphasized that cracks and voids constitute two principal forms of damage to concrete structures, posing risks of reduced structural load-bearing capacity and

impermeability. These issues can ultimately lead to structural failure and catastrophic consequences. Excessive and uncontrolled cracking of structural elements may exacerbate corrosion and compromise the integrity of embedded steel reinforcement. Chun et al. (2021) [34] introduced an automated crack detection approach predicated on image processing techniques, notably leveraging an optical gradient intensifier and a supervised machine learning (ML) method. The method underscored the importance of identifying pertinent features within supervised ML to achieve precision in results. Consideration was given to factors such as the value of target pixels and geometric characteristics characterizing cracks via linear connections. The findings attested that the proposed method achieved an accuracy rate of 99.7% when applied to photographs of concrete structures under challenging conditions, including the presence of shadows and contaminants. Kim et al. (2021) [35] devised a strategy for crack identification by amalgamating RGB-D technology with high-resolution digital cameras, facilitating accurate crack measurements independent of viewing angles. The camera system featured a custom sensor fusion algorithm dedicated to crack identification, ensuring high measurement resolution and robust depth estimation, effectively addressing deflection angles.

Furthermore, regarding the issue of temperature effects on basement concrete structures, there are numerous research cases. Huang et al. (2018) [36], based on a 1:5 scale model of an arch bridge pier, measured the temperature field and temperature history caused by the hydration heat of core concrete. They also proposed an effective heat control method, namely the cold water pipe cooling method. Through thermal stress analysis, the results indicated that the pipe cooling method is an effective approach for reducing hydration temperature and thermal stress. Xin et al. (2018) [37], through experimental research, investigated the influence of temperature history and restraint level on the early performance of concrete, including cracking temperature, cracking stress/strength, creep/free deformation, and cracking potential. The results demonstrated significant differences in the cracking potential of concrete under different temperature histories. Toktorbai uulu et al. (2021) [38] estimated the thermal parameters of various types of concrete and studied their impact on the surface temperature of high-temperature pavement. They sought effective strategies to reduce the surface temperature of asphalt concrete in tunnels through different substrate boundary conditions and the effects of water spraying. Asphaug et al. (2021) [39] pointed out that humid heat simulation is widely used to predict and optimize the hygrothermal performance of building envelope structures. However, determining variations in external hygrothermal boundary conditions for underground walls and floors is challenging due to the diversity and complexity of thermal and moisture loads and the large surface area required for simulating the surrounding ground. Khurshid et al. (2022) [40] used simulations to model molten skin, concrete components, heat transfer, and associated chemical reactions. The results indicated significant effects of skin temperature, concrete components, and injection time.

It is evident that despite some achievements in research, current studies still have certain limitations. Researchers collect temperature data for concrete structures through experiments and on-site monitoring. These data are used to validate numerical simulation results and understand the actual impact of temperature effects on structures. Temperature control techniques include the use of cooling agents, temperature sensors, and insulation materials to control the temperature distribution of concrete structures, reducing the risk of temperature-induced cracking. However, the morphology and size of basement concrete cracks are complex and diverse, and current crack identification and control methods still require improvement in accuracy and efficiency. The impact of temperature fluctuations on basement concrete structures constitutes a multifaceted and pivotal factor, exerting a direct influence on structural stress, deformation, and, by extension, the initiation and propagation of cracks. However, prevailing research on temperature effects predominantly relies on theoretical analyses and

empirical formulas, warranting a more comprehensive approach involving systematic numerical simulations and optimization methodologies. This investigation employs the advanced Mask-RCNN algorithm to undertake the segmentation of crack images within basement concrete structures, enabling precise localization and characterization of cracks. In comparison to conventional image processing techniques, the Mask-RCNN algorithm exhibits superior accuracy and stability. It serves as an effective remedy for the challenges inherent in crack identification, thus bridging extant research gaps in this domain.

## 3 Research methodology

### 3.1 Application of the Mask-RCNN algorithm in crack identification

The Mask-RCNN represents an amalgamation of object detection and image segmentation methodologies founded on CNNs. Its fundamental concept entails the integration of these two tasks to achieve precision at the pixel level [41, 42]. Through the employment of the Mask-RCNN algorithm, the image of the basement concrete structure can be meticulously segmented during the crack identification process, thereby facilitating the precise determination of each crack's spatial location and morphology. Consequently, this furnishes highly accurate input data requisite for subsequent optimization endeavors aimed at crack control. The Mask-RCNN model comprises three principal components: the Backbone Network, the Region Proposal Network (RPN), and the Mask Branch, collectively constituting its structural framework, as illustrated in Fig 1:

Fig 1 shows that the backbone network uses pre-trained CNNs, where VGGNet is used to extract image features. Via a sequence of multi-layered convolutional and pooling operations, the Backbone Network systematically diminishes the dimensions of feature maps while concurrently extracting more advanced semantic information. The RPN assumes responsibility for the generation of candidate bounding boxes intended for target encapsulation. This is achieved by employing a small window to traverse the feature map derived from the Backbone Network, subsequently executing binary classification and regression operations (pertaining to bounding box coordinate adjustments) at each window location. By imposing predefined thresholds and implementing non-maximum suppression, the RPN assembles a set of candidate boxes that have the potential to encompass targets. Conversely, the Mask Branch is tasked with the creation of pixel-level masks for each target. For every candidate box, the Mask Branch takes into account its corresponding feature map region as input. A progression of convolution and upsampling operations within the Mask Branch gradually restores the mask dimensions to align with those of the target. The ultimate outcome is a binary target mask achieved through threshold processing.

In order to enhance the precision and robustness of crack identification utilizing the Mask-RCNN algorithm on basement concrete structure images, a series of preprocessing steps are implemented. These steps encompass image data preprocessing, the annotation of crack regions, and the generation of training and validation samples, which are carried out as follows:

1. Image Enhancement: Firstly, the experiment applied various image enhancement techniques, including brightness adjustment, contrast enhancement, and histogram equalization. These techniques contribute to improving the visual quality and contrast of the images, making the crack areas more distinct and prominent. By adjusting brightness and contrast, this research was able to capture finer details and the texture of cracks in the images, providing better input images for subsequent crack detection.

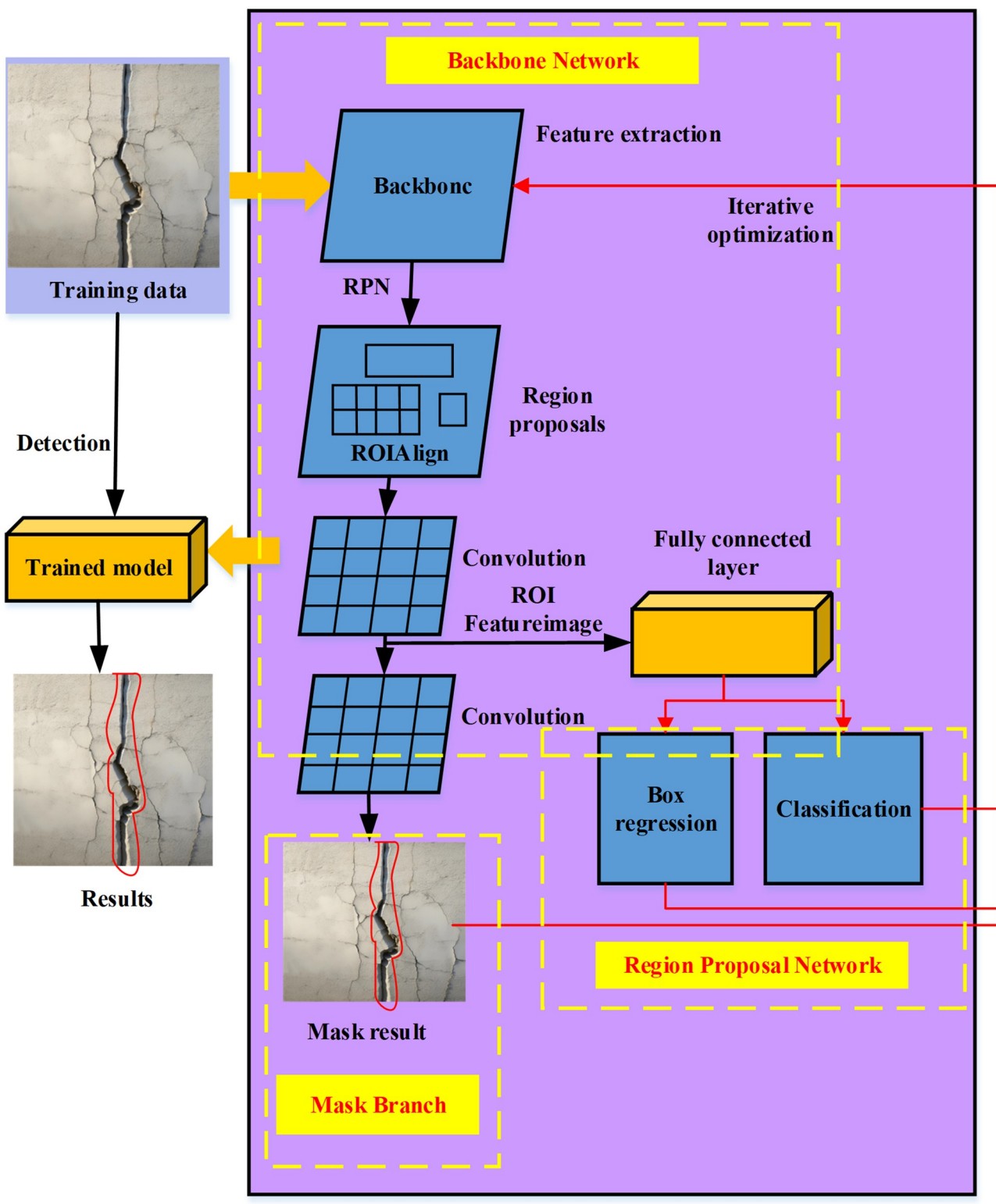

**Fig 1. The framework structure of the Mask R-CNN.**

2. Resizing: Basement concrete structure images often have different dimensions. Therefore, the experiment performed resizing to ensure that the images had consistent dimensions. This step involved maintaining the aspect ratio and resizing the images to a fixed size or adjusting them to the same dimensions through cropping and padding operations. This helps ensure that the input images received by the neural network model have uniform dimensions, enabling the model to process images of different sizes.

3. Image Normalization: Finally, the experiment normalized the images by scaling pixel values to a fixed range of [–1, 1]. This step aids in training and optimizing the neural network model, as it ensures that the data have similar numerical ranges, avoiding training issues such as gradient vanishing or exploding. By normalizing the pixel values of the images to a standard range, this research was able to maintain the stability and convergence of the model.

The objective of these preprocessing steps is to enhance the quality of the input images, making them more suitable for training the Mask-RCNN model and crack recognition. In Fig 2, this research provides a visual comparison of basement concrete structure images before and after preprocessing to demonstrate the effects of the preprocessing steps. These improvements can assist the model in better identifying and locating cracks, thereby enhancing the accuracy and robustness of crack detection.

Fig 2 shows the changes in the image after each preprocessing step. Subsequently, the identified crack regions undergo a process of annotation, yielding the essential training and

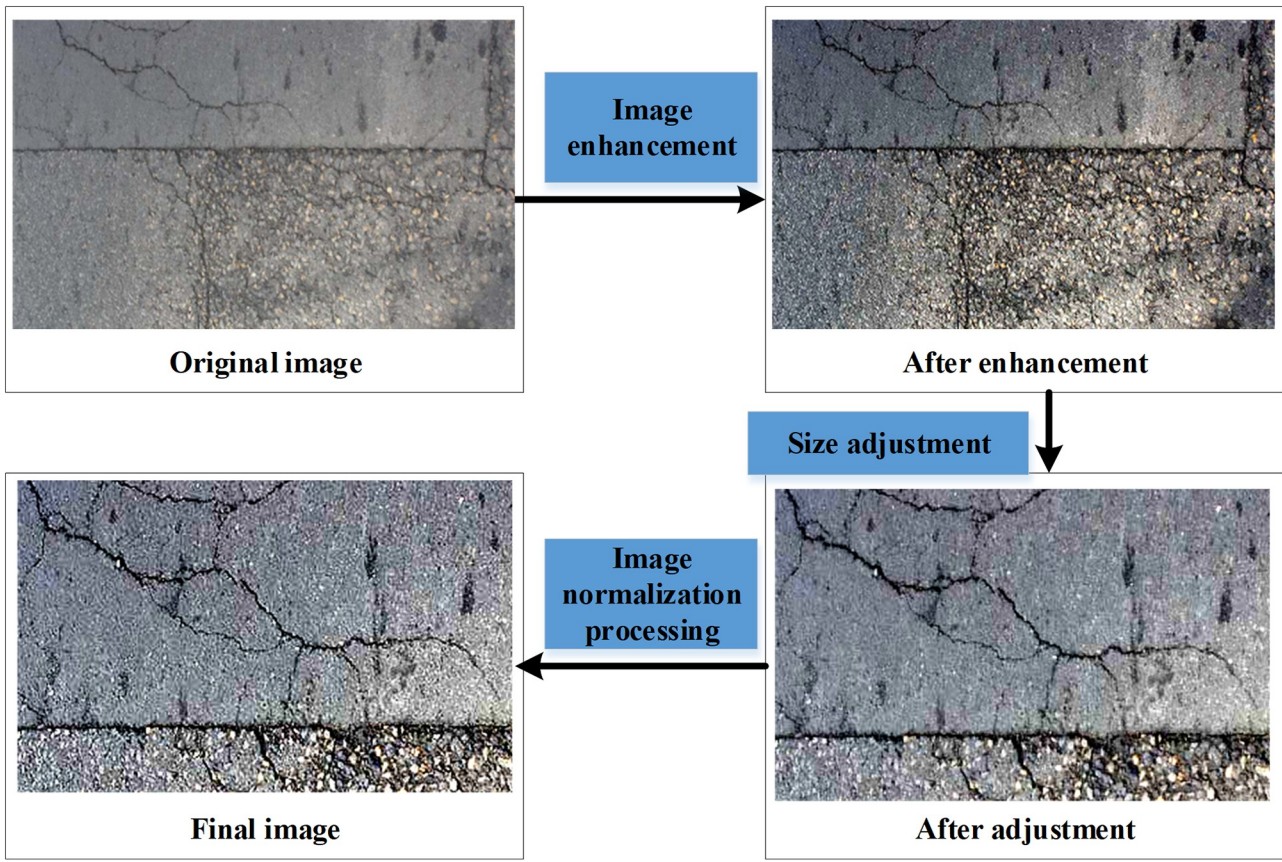

**Fig 2. Comparison of the effects before and after image preprocessing.**

validation samples that serve as instructive data for the Mask-RCNN algorithm's learning and crack identification. This process unfolds in two distinct stages: Firstly, experts manually delineate the basement concrete structure images, meticulously demarcating the precise positions and configurations of the cracks. This meticulous task is executed with the assistance of specialized image annotation tools and graphic design software. Secondly, predicated on the outcomes of the manual annotation, corresponding label images are meticulously generated. These labeled images mirror the dimensions of the original images, designating the cracked and non-cracked regions as the foreground (representing positive samples) and background (representing negative samples).

Following the annotation phase, the Mask-RCNN model is employed for the purpose of detecting cracks within the basement concrete structure. The model undergoes supervised learning, where its loss function encompasses three integral components: the neural network classification loss denoted as $L_{cls}$, the bounding box loss denoted as $L_{box}$, and the mask loss denoted as $L_{mask}$. The classification loss serves to quantify the model's accuracy in distinguishing between cracks and non-cracks. Meanwhile, the bounding box loss gauges the model's precision in pinpointing the shape and location of the cracks. Lastly, the mask loss assesses the model's pixel-level segmentation accuracy with respect to the crack regions, a metric expressed through Eq (1):

$$L = L_{cls} + L_{box} + L_{mask} \tag{1}$$

In the realm of supervised learning, the model undergoes a training regimen that hinges upon labeled crack image data. This training dataset comprises comprehensive information encompassing the precise location and morphology of the cracks alongside their corresponding crack masks. Capitalizing on this dataset as input, the model parameters undergo continuous refinement facilitated by the iterative processes of forward propagation and backpropagation. These iterative adjustments fine-tune the model's capacity to make more precise predictions regarding the cracks' location, configuration, and mask. Over the course of this iterative training trajectory, the model progressively acquires a nuanced understanding of the inherent characteristics of cracks, thereby enhancing its proficiency in the task of crack identification. In order to mitigate the risk of overfitting, dropout regularization techniques are strategically employed, bolstering the model's generalization capabilities. This safeguard ensures that the model's performance extends beyond the constraints of the training dataset. The orchestrated workflow, depicted in Fig 3, delineates the Docker's operational processes:

When employing the Mask R-CNN model to train data related to concrete cracks, the algorithmic process is depicted in Fig 4:

Fig 4 portrays a systematic sequence of operations. Initially, a crack map of the basement concrete structure serves as the input, and the image undergoes a series of preprocessing steps, which encompass resizing and normalization. Following this preprocessing, the pre-trained neural network engages in feature map extraction from the image. A predetermined number of candidate boxes are subsequently generated for each pixel within the feature map. These candidate boxes represent regions that may potentially encompass cracks of interest. Subsequently, the candidate boxes undergo boundary box regression and binary classification via the candidate box generation network. This step serves to screen and filter the candidate boxes, retaining those that exhibit the most salient crack characteristics. The filtered candidate frame is subjected to ROIAlign operation. This operation establishes a precise correspondence between the original image and the feature map, aligning each region within the feature map with its corresponding area in the original image. Lastly, a Fully Connected Network (FCN) comes into play, enabling the classification of each candidate box into N categories, boundary

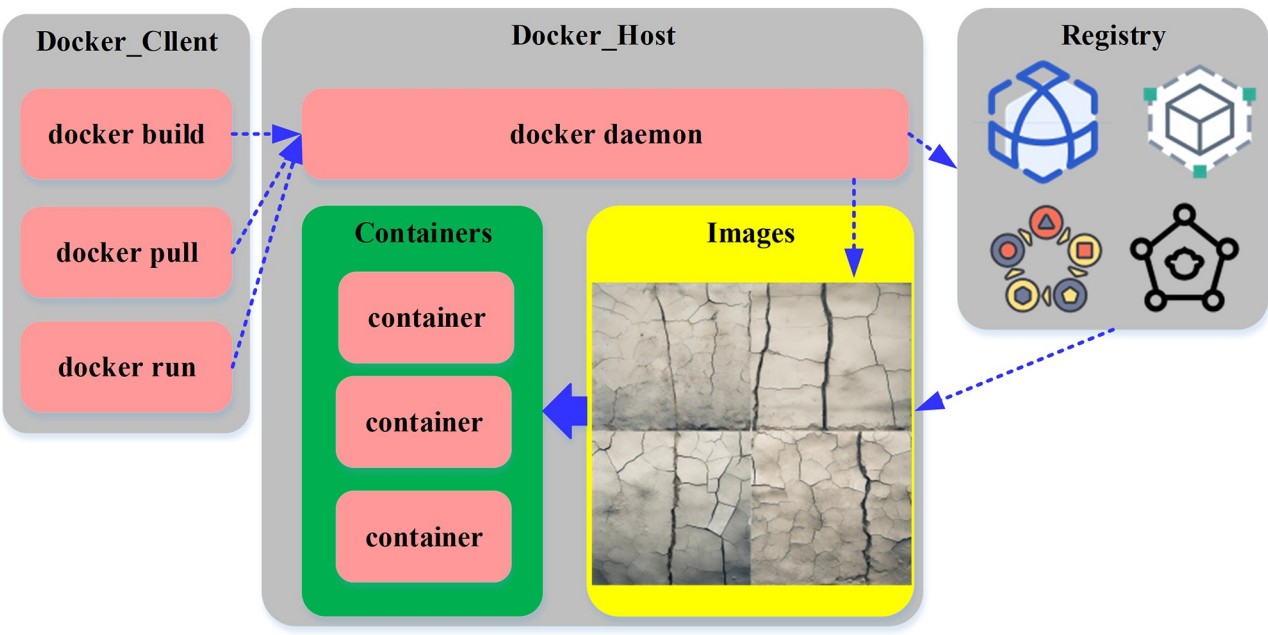

**Fig 3. The workflow of Docker.**

box regression, and mask generation. This pivotal stage encompasses the classification of each candidate box, discerning between cracks and non-cracks, predicting the precise location and configuration of the cracks, and ultimately generating a mask to represent the identified cracks.

### 3.2 Simulation of stress and deformation of structures under temperature changes based on FEA

In parallel, the FEA method is deployed to simulate the stress and deformation characteristics of the basement concrete structure under the influence of temperature fluctuations. FEA stands as a numerical simulation technique that resolves the mechanical behavior of a structure through the subdivision of said structure into a finite number of units, subsequently establishing the relationships between these units [43]. The initial phase of this process entails the creation of a 3D model that faithfully replicates the geometry and dimensions of the actual basement concrete structure. The structure is discretized into a finite number of units, often in the form of triangular or quadrilateral elements, in an effort to approximate the structural geometry effectively. Subsequently, the structure undergoes meshing, whereby the structural units are subdivided into a series of smaller subunits. The selection of grid density and cell size is a crucial consideration and must be determined judiciously in accordance with the structural complexity and the specific analysis requirements to ensure the precision of the analysis results.

Within the realm of FEA, a damage model takes center stage to elucidate the mechanical behavior of the concrete. Notably, the Idealized Elastic-Plastic Damage Model, a commonly employed damage plastic model, is chosen for this purpose. This model encapsulates the interplay of elastic deformation, damage accumulation, and plastic deformation within concrete under the influence of stress. The model's mathematical formulation is articulated as shown in

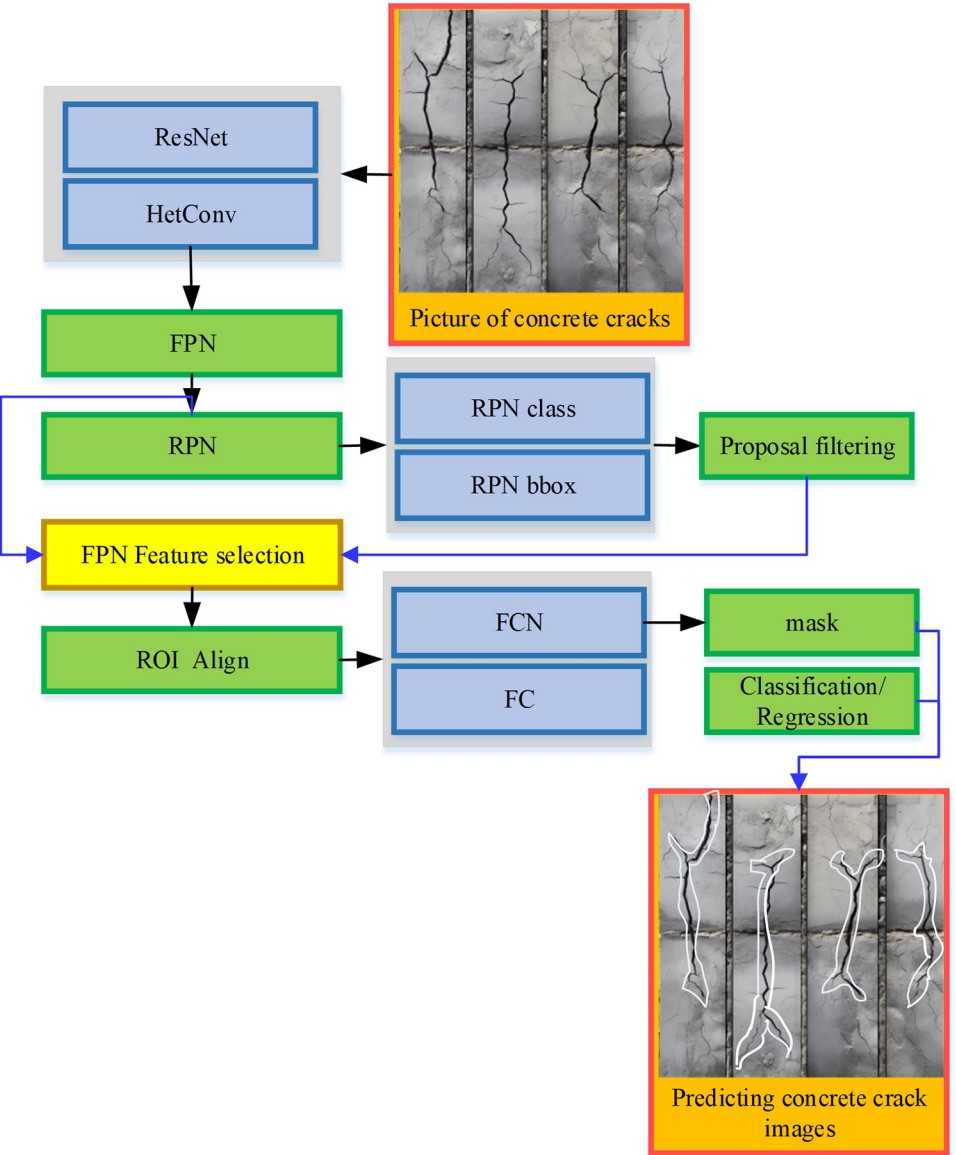

**Fig 4. The framework structure of the Mask R-CNN.**

Eq (2):

$$\sigma = (1 - D) \times E \times \varepsilon \tag{2}$$

In Eq (3), the symbols used are as follows: $\sigma$ denotes the stress experienced by the concrete material. D represents the damage parameter. E signifies the elastic modulus. $\varepsilon$ corresponds to the strain experienced by the concrete. Eq (3) facilitates the computation of the damage parameter:

$$D = 1 - (1 - \varepsilon/\varepsilon_0)_b \tag{3}$$

In Eq (3) $\varepsilon_0$ denotes the strain threshold that initiates damage, and b represents the material parameter associated with damage. This damage model encompasses considerations of elastic

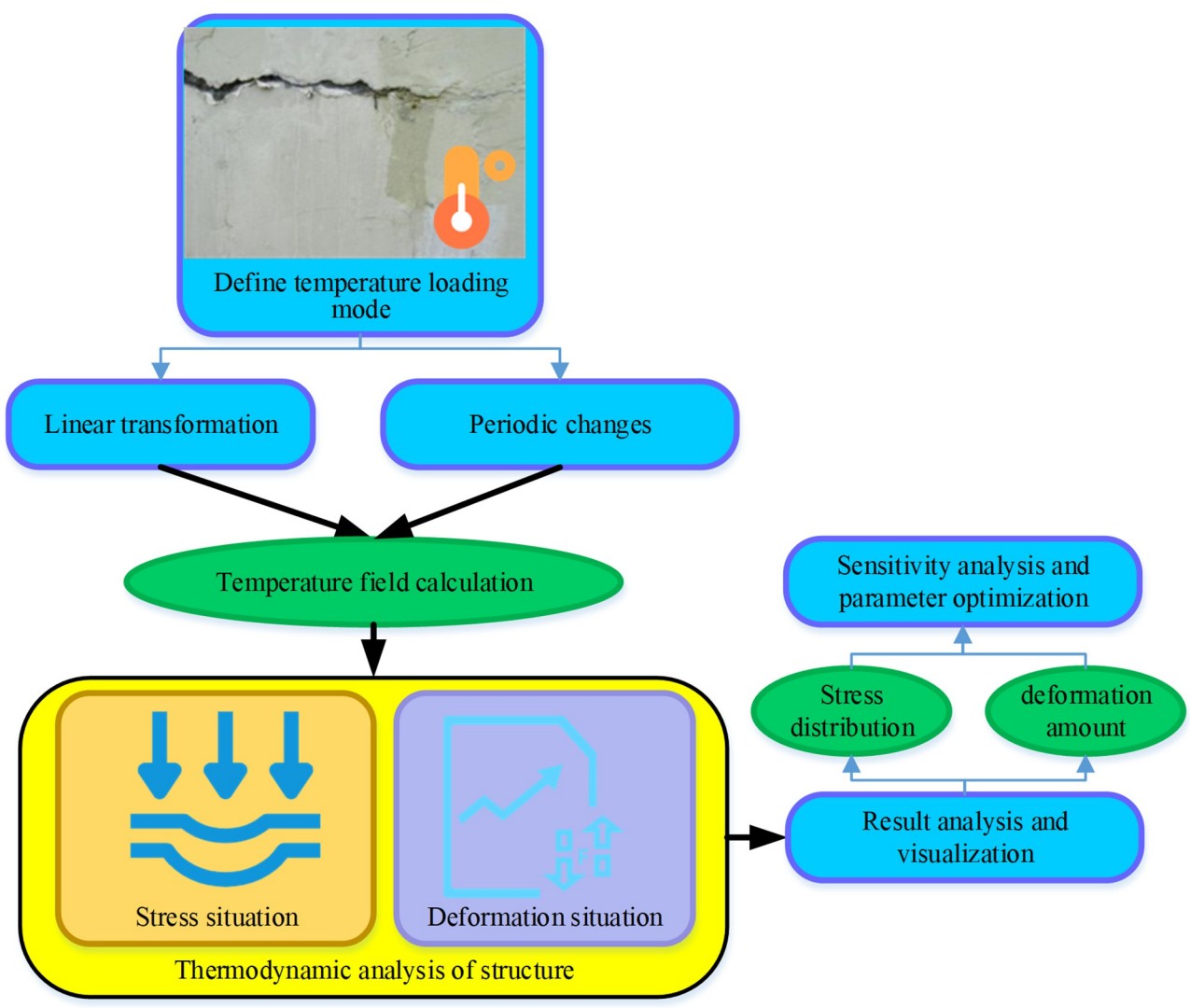

**Fig 5. The procedures of the temperature effect analysis.**

deformation, plastic deformation, and the accumulation of damage within the concrete material during the stress-induced processes. The analysis of temperature effects is predicated upon the established finite element framework and the delineated material models. A comprehensive outline of the specific procedural steps is presented in Fig 5:

Numerous factors must be taken into account when addressing the effects of temperature. The seasonal temperature differential experienced by the basement wall can be expressed as presented in Eq (4):

$$\Delta T(t, t_0) = 0.5(T_{max} - T_{min}) \times \left[\frac{cos2n\pi(t - t_\lambda)}{365} - \frac{cos2n\pi(t_0 - t_\lambda)}{365}\right] \tag{4}$$

In Eq (4), $T_{max}$ denotes the maximum annual temperature encountered by the structure, $T_{min}$ represents the minimum annual temperature and $t_\lambda$ signifies the time of highest environmental occurrence. A pivotal factor influencing the temperature-induced stress in concrete is

the heat generated during cement hydration, a parameter closely linked to the concrete's age. This relationship is defined by Eq (5):

$$Q(\tau) = Q_u(1 - e^{-m\pi})$$ (5)

In Eq (5), Q($\tau$) signifies the total heat of hydration at age $\tau$, expressed in kJ/kg. $Q_u$ represents the ultimate heat of hydration, while m is a constant determined by various factors, including cement type, casting temperature, and specific surface area.

$$\theta(\tau) = \frac{Q(\tau)(W + kF)}{c\rho}$$ (6)

The calculation procedure for the temperature rise in insulated concrete involves the following components: the quantity of cement used ($W$), the specific heat of cement ($c$), concrete density ($\rho$), the quantity of mixed material ($F$), the heat of cement hydration ($Q(t)$), and the reduction factor ($k$), which considers factors like fly ash (k = 0.25).

Concrete shrinkage ranks among the foremost contributors to concrete cracking. The dry shrinkage strain value of concrete at any given age, $\varepsilon_{sh}(t)$, can be calculated according to European recommended standards:

$$\varepsilon_{sh}(t) = \varepsilon_c K_b K_t K_e$$ (7)

In Eq (7), $\varepsilon_c$ represents the shrinkage strain value determined from the relative humidity-shrinkage relationship diagram, $K_b$ stands for the fitting ratio coefficient, $K_t$ denotes the drying time correction factor and $K_e$ represents the effective thickness influence factor.

## 3.3 Design and implementation of the crack control optimization algorithm

In the algorithm aimed at optimizing crack control, it is imperative to define a suitable objective function for guiding the adjustment of geometric parameters and material properties associated with basement concrete structures. The optimization function's primary objective is to mitigate stress concentration, deformation, crack density, as well as crack length and width. The formulation of the objective function is expressed as shown in Eq (8):

$$J = w_1 \times S + w_2 \times D + w_3 \times L + w_4 \times W$$ (8)

In Eq (8), J represents the value of the objective function. S signifies the degree of stress concentration. D indicates the amount of deformation. L stands for the crack length. W represents the crack width. $w_1$, $w_2$, $w_3$, and $w_4$ represent the respective weighting coefficients that balance the influence of various factors on the optimization objective.

In order to optimize crack control effectively, the selection of an appropriate optimization algorithm is essential for searching for the optimal solution. In this context, a genetic algorithm is employed for optimal control. Initial parameter settings of the optimization algorithm, such as population size, crossover rate, mutation rate, and others, are adjusted to enhance the convergence speed and search effectiveness of the optimization process. The overarching goal of the optimization is to minimize f(x), which represents the total length of cracks present in the basement concrete structure, while simultaneously adhering to the following

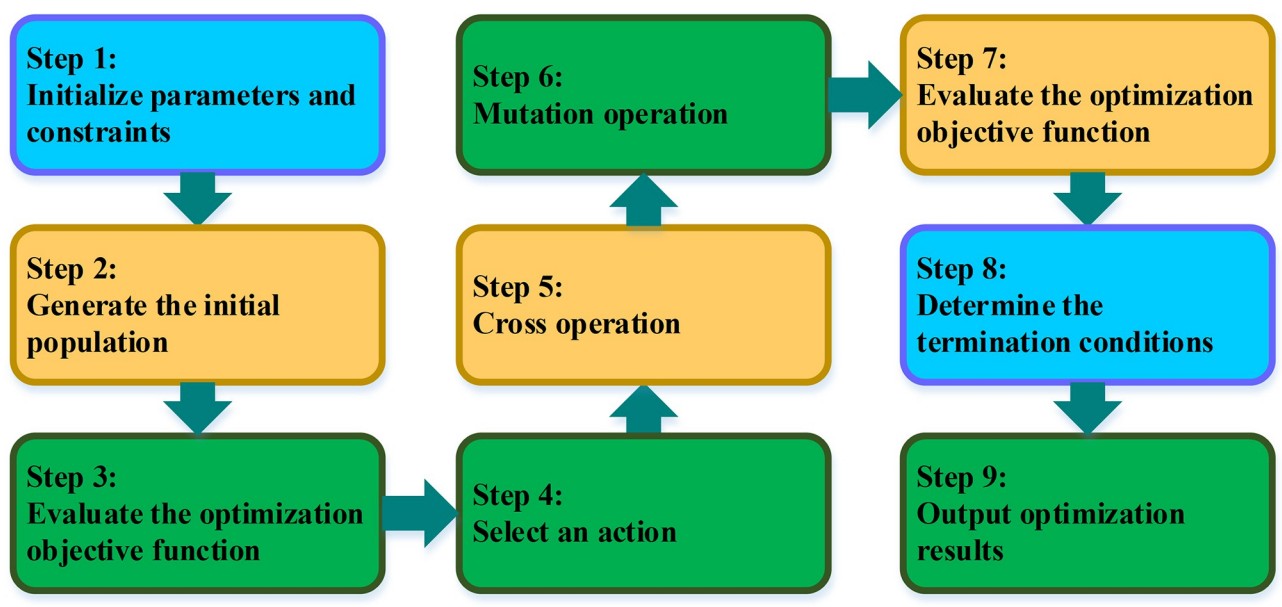

**Fig 6. The crack control optimization algorithm.**

constraints:

$$\sigma \leq \sigma_{max} \tag{9}$$

$$\delta \leq \delta_{max} \tag{10}$$

$$lb \leq x \leq ub \tag{11}$$

In Eqs (9)–(11). $\delta$ denotes the deformation experienced by the structure. $\delta_{max}$ representing the predetermined deformation limit. The variable x signifies the parameter values under consideration. $lb$ and $ub$ are indicative of the lower and upper bounds for these parameters. It is imperative to ensure that the maximum stress $\sigma$ remains within the structural material's load-bearing capacity, denoted as $\sigma_{max}$, and that the structure does not surpass the preset maximum deformation limit. Therefore, the parameter values adjusted during the optimization process must fall within a reasonable and acceptable range. The specific implementation procedure of the crack control optimization algorithm is detailed in Fig 6:

### 3.4 Experimental procedure

The subject of this research is the basement concrete structure of a prominent urban building selected as the research object. Table 1 provides an overview of the structural parameters.

These parameters will serve as the input data for the experimental process involving FEA and optimization algorithms. The goal is to assess the stress, deformation, and crack development within the basement concrete structure under varying temperature conditions. The experimentation was conducted on five separate occasions, each time with different temperature conditions. Real-time measurements of the structure's stress and deformation were obtained through sensors and measurement equipment. Additionally, relevant performance parameters of the concrete materials employed in the structure, encompassing elastic modulus

**Table 1. The parameters of the basement concrete structure.**

| Parameter | Value |
|---|---|
| Basement name | Basement A |
| Basement size | Length: 20m, width: 10m, height: 5m |
| Concrete strength of basements | Compression strength: 30 MPa |
| Concrete materials for basements | Elastic modulus: 30 GPa |
| Temperature conditions | Initial temperature: 25˚C, maximum temperature: 50˚C |

and tensile strength, were recorded. Detailed environmental parameter configurations are presented in Table 2:

Evaluation Criteria: 1) Crack Density: The analysis involves comparing the variation in crack density before and after optimization. This is accomplished through statistical analysis, examining the number and distribution of cracks observed during the experiment. 2) Average Crack Length: Measurement and calculation of crack lengths observed during the experiment are conducted, followed by a comparison of the change in average crack length before and after optimization. 3) Average Crack Width: Crack widths observed during the experiment are measured and calculated, allowing for a comparison of the variation in average crack widths before and after optimization. 4) Stress Concentration Level: The assessment of stress concentration within the structure, both before and after optimization, is performed using FEA or other stress analysis methods. 5) Deformation: In the experimental phase, structural deformation is measured and calculated. A comparison is then made to evaluate the variation in maximum deformation before and after optimization.

## 4 Results and discussion

### 4.1 Crack identification and display of analysis results

Fig 7 illustrates the maximum stress and deformation observed under various experimental conditions:

In Fig 7, as the temperature increases from Experiment 1 (25˚C) to Experiment 5 (45˚C), the maximum stress gradually decreases, reducing from 120 MPa to 100 MPa. This data indicates that with rising temperatures, the stress level within the structure decreases, possibly due to the weakening of material strength properties at higher temperatures. The maximum deformation exhibits slight fluctuations under different experimental conditions but generally remains at a lower level, ranging from approximately 3.2 mm to 3.8 mm. This suggests that the deformation of the structure under varying temperature conditions is controlled within acceptable limits, maintaining the stability and safety of the structure. This implies that the optimization algorithm effectively reduces the structure's stress levels under different temperature conditions, consequently reducing structural loading and stress concentration

**Table 2. Setting of experimental environment parameters.**

| Experiment number | Temperature (˚C) | Humidity (%) | Loading method | Time (hours) |
|---|---|---|---|---|
| Experiment 1 | 25 | 50 | Constant temperature loading | 24 |
| Experiment 2 | 30 | 60 | Temperature change loading | 48 |
| Experiment 3 | 35 | 55 | Constant temperature loading | 72 |
| Experiment 4 | 40 | 45 | Temperature change loading | 36 |
| Experiment 5 | 45 | 50 | Constant temperature loading | 48 |

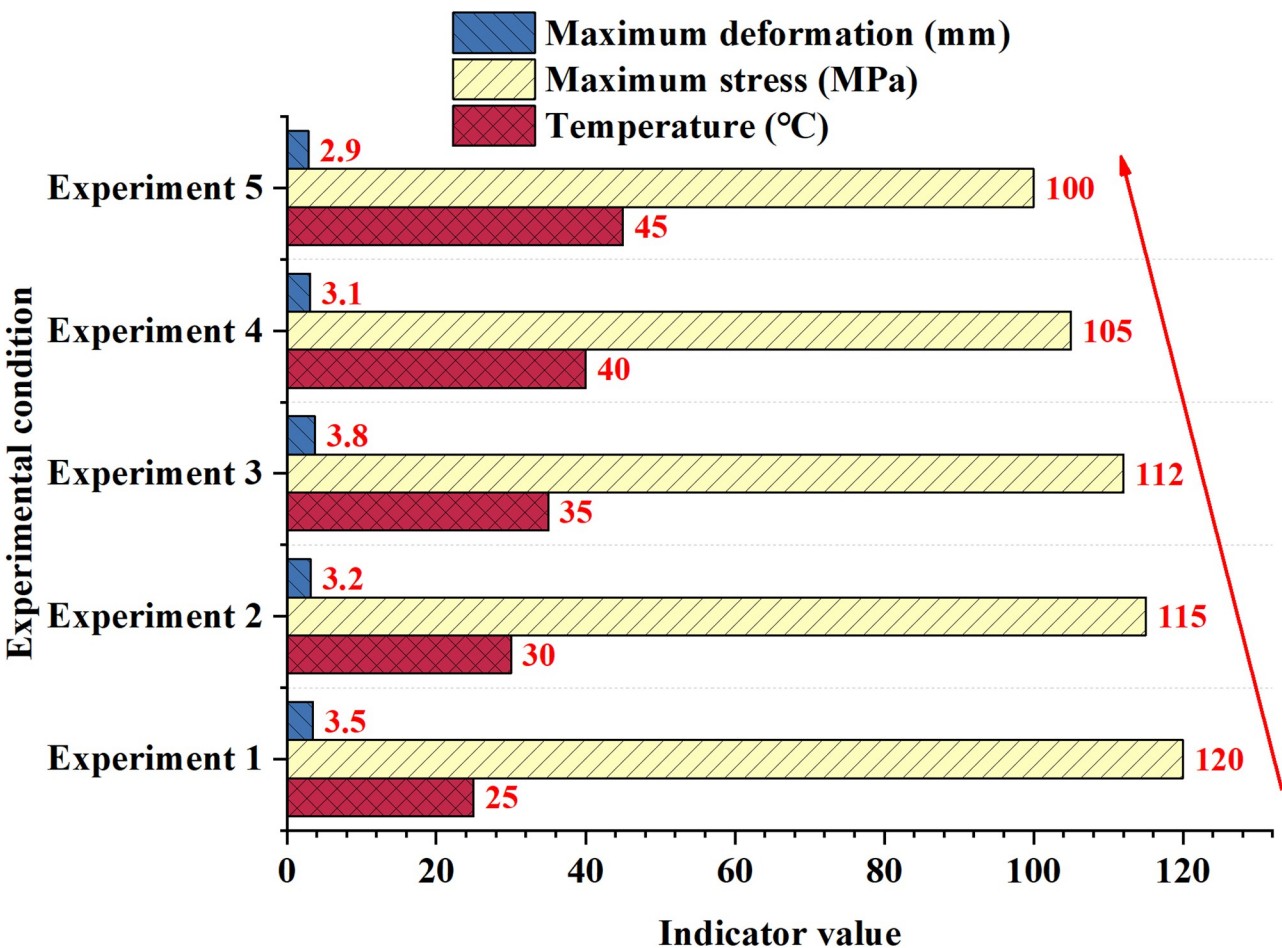

**Fig 7. Results of stress and deformation analysis in diverse experiments.**

phenomena. This is crucial for the durability and safety of the structure. Furthermore, although the maximum deformation varies with different temperatures, it remains within acceptable ranges, indicating that the optimization algorithm also performs well in controlling structural deformation. This contributes to maintaining the stability of the structure, ensuring its normal operation under varying temperature conditions. The application of the optimization algorithm in this research appears to effectively enhance the performance of the concrete structure under different temperature conditions. It reduces the maximum stress levels, lowering structural risks and keeping structural deformations within acceptable ranges. This holds significant implications for the design and maintenance of basement concrete structures, enhancing their stability and reliability. Fig 8 presents the outcomes of crack identification and analysis across a range of temperature conditions:

Fig 8 illustrates a noticeable trend as the temperature rises. There is a gradual reduction in crack density, decreasing from 0.032 to 0.018. Simultaneously, both average crack length and width exhibit a gradual reduction, diminishing from 15.4mm to 9.2mm. The stress concentration degree also experiences a gradual decline, decreasing from 0.12 to 0.08, while the deformation of the structure decreases at a slower rate. This observation signifies the optimization algorithm's effectiveness in reducing crack density within concrete structures and enhancing their resistance to cracking under elevated temperatures. Furthermore, it effectively curtails

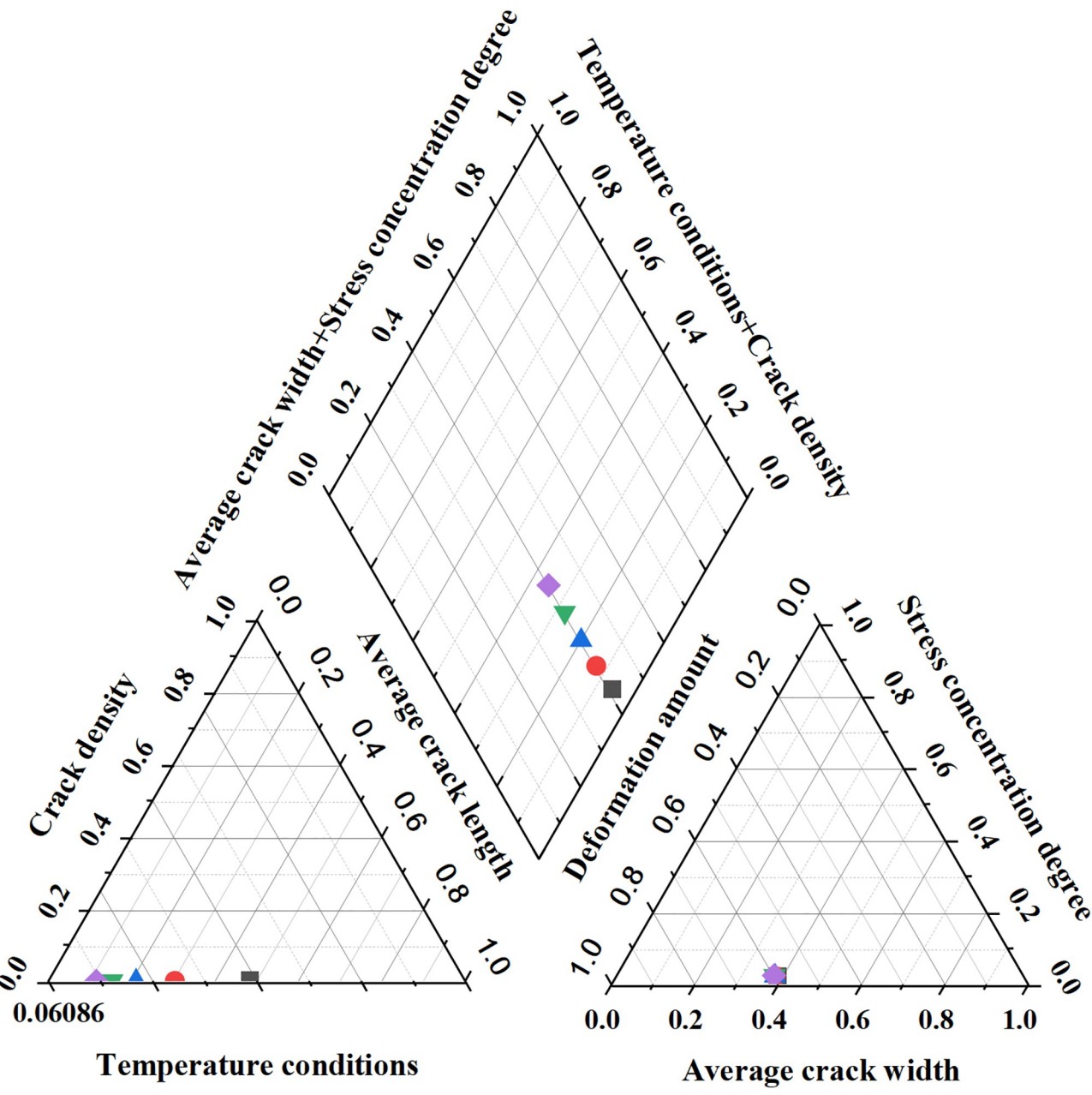

**Fig 8. Crack identification and analysis results.**

the development and expansion of cracks, leading to a notable reduction in their dimensions. Consequently, the proposed crack control optimization algorithm demonstrates its capability to diminish crack density and size across varying temperature conditions. Additionally, it enhances the distribution of stress within the structure and improves deformation performance, thereby bolstering the crack resistance and stability of the basement concrete structure.

The enhancement achieved by the optimization algorithm in reducing cracks is visually depicted in Fig 9:

Fig 9 provides compelling evidence of the substantial reductions achieved in crack density, average crack length, maximum stress concentration, average crack width, and maximum

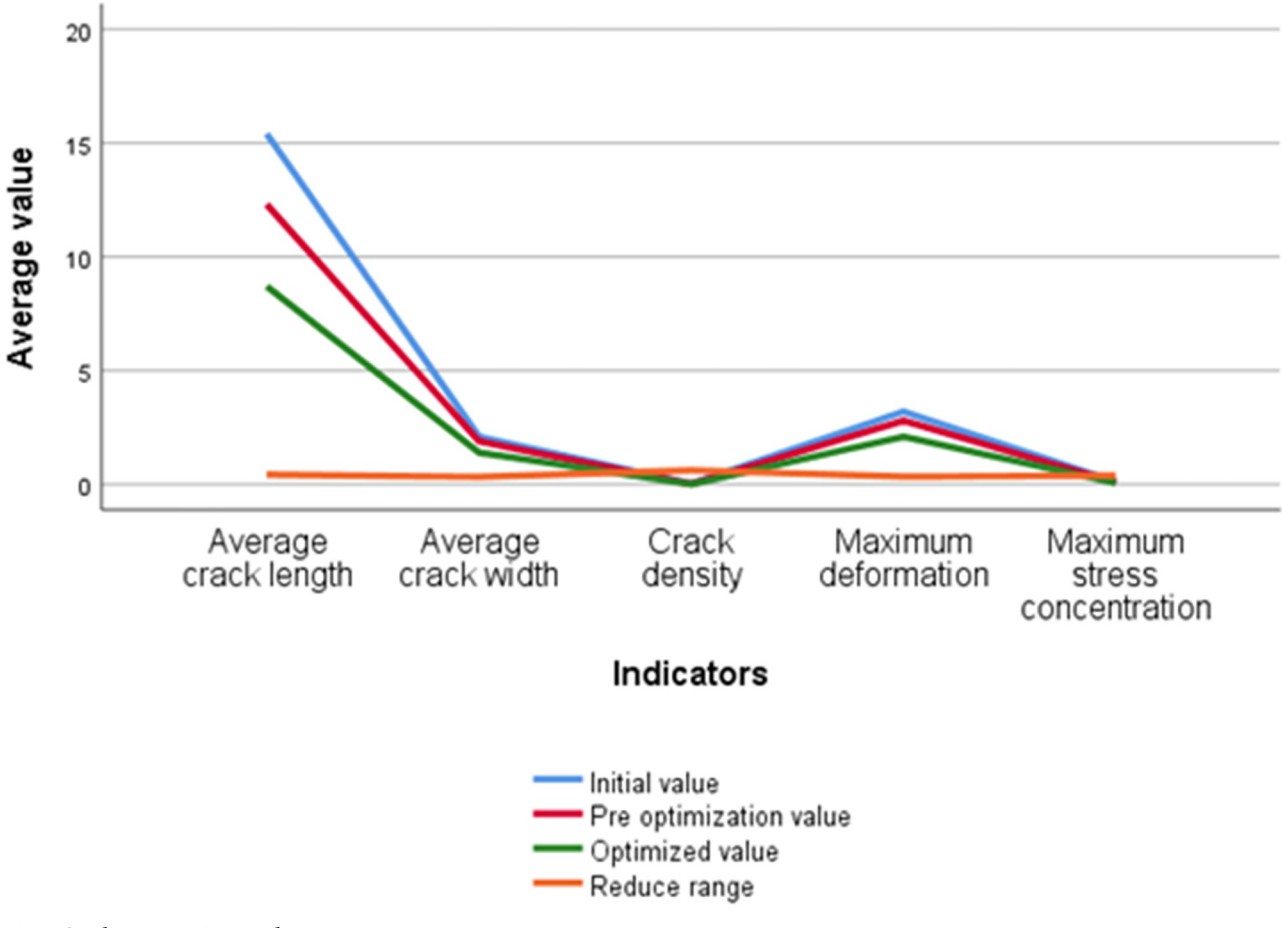

**Fig 9. Crack optimization result.**

deformation through the meticulous adjustment of the structure's geometric parameters and material properties. Notably, the crack density sees a remarkable reduction of 62.5%, while the average crack length and width decrease by 43.48% and 33.33%, respectively. Furthermore, there are significant reductions of 39.17% in maximum stress concentration and 34.37% in deformation. This compelling data underscores the significant advantages offered by the optimization algorithm in effectively mitigating the occurrence and propagation of cracks within the basement concrete structure.

The comparison with other traditional optimization methods is shown in Fig 10:

In Fig 10, the method proposed in this research performs the best in terms of reducing crack density, achieving a remarkable reduction range of 62.5%, far surpassing the other three traditional methods. Canny edge detection and Hough transform methods exhibit relatively lower reductions at 45.2% and 52.2%, respectively. The region-growing algorithm shows slightly lower performance with a reduction of 40.2% in crack density. Similarly, the method proposed in this research also excels in reducing the average crack length, achieving a reduction range of 43.48%. Canny edge detection and Hough transform achieve reductions of 30.5% and 38.5%, respectively. The region-growing algorithm performs the poorest in reducing the average crack length, with only a 25.3% reduction. Furthermore, the proposed model in this research also demonstrates relatively good performance in reducing the average crack width, with a reduction of 33.33%. Canny edge detection and Hough transform achieve reductions of

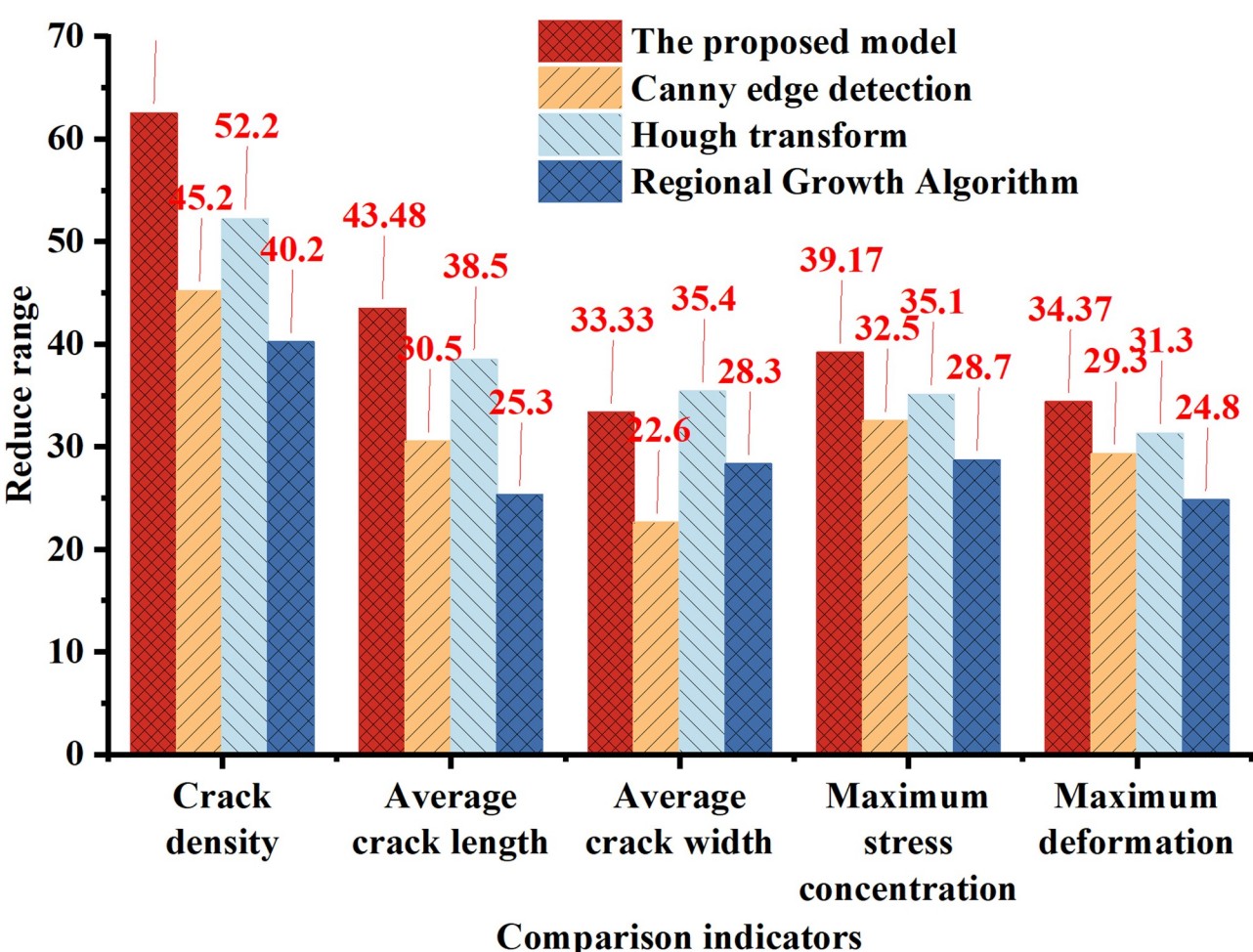

**Fig 10. Comparison of crack optimization results using different image recognition methods.**

22.6% and 35.4%, respectively. The region-growing algorithm shows moderate performance with a reduction of 28.3% in average crack width. Similar results are observed in the reduction ranges of maximum stress concentration and maximum deformation. Overall, the model proposed in this research exhibits a significant advantage over the traditional Canny edge detection, Hough transform, and region-growing algorithm in optimizing concrete cracks. It achieves more pronounced effects in reducing crack density, average crack length, average crack width, maximum stress concentration, and maximum deformation. This result suggests that deep learning methods (Mask-RCNN) hold potential application value in concrete structure crack optimization and may assist the engineering field in better controlling and reducing cracks.

## 4.2 Display of FEA results for temperature effects

The contour lines depicting stress within the concrete wall of the basement typically exhibit a circular distribution, and the impact of varying temperature differentials on temperature-induced stress is illustrated in Fig 11.

In Fig 11, when the temperature differential surpasses 25°C, the tensile strength of the basement concrete proves insufficient to withstand the maximum stress concentrated at the center

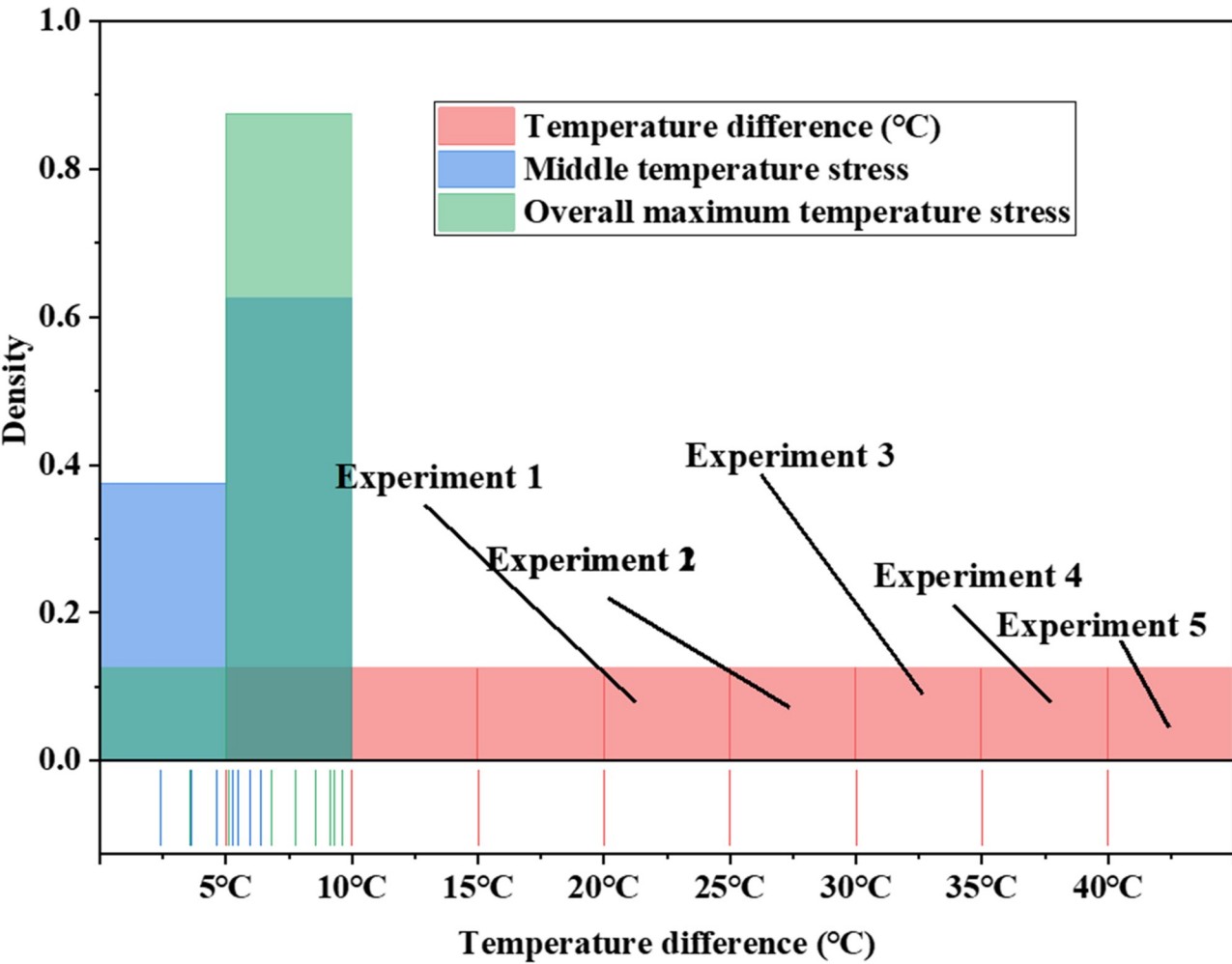

**Fig 11. The effect of different temperature differences on temperature stress.**

of the wall, which represents the overall peak stress. Consequently, regulating the temperature differential along the exterior wall of the basement becomes of paramount importance to mitigate temperature-induced stress. Moving from the wall's two ends toward the center, stress levels gradually decrease, accompanied by an expanding range of stress fluctuations. Examining the vertical dimension of the wall, it becomes evident that stress escalates from top to bottom, with a corresponding increase in stress fluctuation amplitude. This stress distribution pattern is indicative of the foundation's influence on wall behavior, exerting specific control over the wall's response. Notably, the region bearing the maximum stress within the wall remains relatively consistent, often situated in the lower right and lower left corners of the wall. The influence of distinct wall lengths on temperature-induced stress is depicted in Fig 12:

In Fig 12, the numerical values of intermediate temperature stress fluctuate under different wall lengths but do not exhibit a clear trend. This may suggest that variations in wall length have a relatively minor impact on intermediate temperature stress or are influenced by other factors. Similarly, the numerical values of overall maximum temperature stress also fluctuate under different wall lengths without displaying a clear trend. Like intermediate temperature stress, this may indicate that changes in wall length have a relatively small effect on the overall

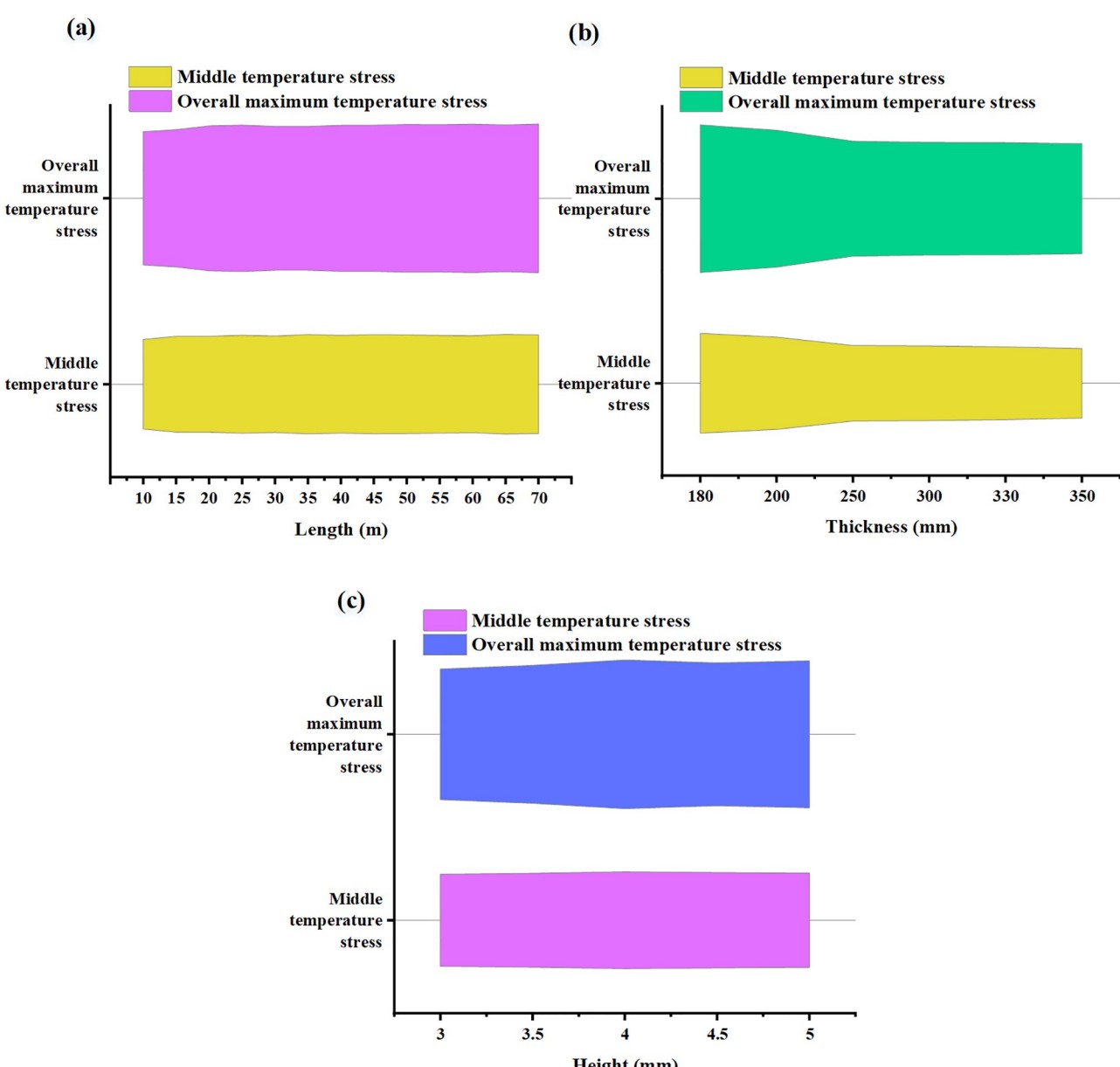

**Fig 12. The influence of different walls on temperature stress (a. Different wall length; b. Different wall thicknesses; c. Different wall heights).**

maximum temperature stress or are influenced by other factors. Additionally, in scenarios where all other parameters remain constant, and the wall thickness measures less than 250mm, both the stress value at the wall's midpoint and the maximum stress value increase proportionally with diminishing wall thickness. This phenomenon arises due to the heightened susceptibility of thinner walls to temperature differentials and their greater degree of constraint by the foundation. The reduced thickness of the wall enhances its thermal conductivity, leading to a concentration of temperature-induced stress within the wall. Nevertheless, when the wall thickness surpasses 250mm, there is no substantial alteration in the maximum temperature-induced stress within the wall or the temperature-induced stress at its midpoint. Consequently, increasing wall thickness exhibits negligible influence on temperature-induced stress.

Furthermore, as the height of the wall increases, the temperature-induced stress within the wall also augments, albeit with a diminishing rate of increase.

## 5 Conclusion

The present study introduces a crack control methodology founded on FEA and an optimization algorithm, enabling precise assessment of cracks within concrete structures and effective reduction of crack lengths through parameter adjustments and optimization. The analysis encompasses the stress and deformation experienced by concrete structures in response to temperature fluctuations, thereby delivering valuable insights for structural design and construction practices. Furthermore, the investigation delves into the influence of wall thickness on temperature-induced stress, elucidating the relationship between wall thickness and temperature-induced stress and offering guidance for the optimal design of wall structures. The principal findings can be summarized as follows: Crack identification and analysis techniques accurately pinpoint and assess cracks within concrete structures, laying a crucial foundation for crack mitigation and maintenance efforts. The results of structural stress and deformation analysis elucidate the mechanical behavior of concrete structures subjected to temperature variations. Implementation of the optimization algorithm effectively reduces the cumulative length of cracks while simultaneously satisfying the structural strength and deformation prerequisites. Nonetheless, certain limitations are worth noting, including the reliance on simulated and analyzed experimental data grounded in specific assumptions that may not perfectly mirror real-world conditions. Consequently, further validation and adjustment of model parameters are planned, coupled with the refinement of the crack control algorithm to encompass a broader spectrum of optimization objectives and constraints, ultimately leading to enhanced crack control outcomes. Subsequently, practical engineering cases will serve as a basis for verification and parameter adjustment, thereby enhancing the accuracy and reliability of simulation results.

## Supporting information

**S1 Data.**
(ZIP)

## Author Contributions

**Conceptualization:** Shouyan Wu.

**Data curation:** Shouyan Wu.

**Formal analysis:** Shouyan Wu.

**Investigation:** Shouyan Wu.

**Methodology:** Shouyan Wu.

**Project administration:** Shouyan Wu.

**Resources:** Shouyan Wu.

**Supervision:** Feng Fu.

**Validation:** Feng Fu.

**Visualization:** Feng Fu.

**Writing – original draft:** Shouyan Wu, Feng Fu.

Writing – review & editing: Feng Fu.

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
