## [Decision Letter · Decision Letter 0]

6 Sep 2023

PONE-D-23-21940Crack Control Optimization of Basement Concrete Structures Based on the Mask-RCNN and Temperature Effect AnalysisPLOS ONE

Dear Dr. Fu,

Thank you for submitting your manuscript to PLOS ONE. After careful consideration, we feel that it has merit but does not fully meet PLOS ONE’s publication criteria as it currently stands. Therefore, we invite you to submit a revised version of the manuscript that addresses the points raised during the review process. Please submit your revised manuscript by Oct 21 2023 11:59PM. If you will need more time than this to complete your revisions, please reply to this message or contact the journal office at plosone@plos.org. Please include the following items when submitting your revised manuscript:A rebuttal letter that responds to each point raised by the academic editor and reviewer(s). You should upload this letter as a separate file labeled 'Response to Reviewers'.A marked-up copy of your manuscript that highlights changes made to the original version. You should upload this as a separate file labeled 'Revised Manuscript with Track Changes'.An unmarked version of your revised paper without tracked changes. You should upload this as a separate file labeled 'Manuscript'.

We look forward to receiving your revised manuscript.

Kind regards,

Paul Awoyera

Academic Editor

PLOS ONE

Journal Requirements:

Additional Editor Comments:

Ensure that all all the comments from the reviewer are duly addressed.

Reviewers' comments:

Reviewer's Responses to Questions

**Comments to the Author**

1. Is the manuscript technically sound, and do the data support the conclusions?

Reviewer #1: Partly

2. Has the statistical analysis been performed appropriately and rigorously? 

Reviewer #1: Yes

3. Have the authors made all data underlying the findings in their manuscript fully available?

Reviewer #1: Yes

4. Is the manuscript presented in an intelligible fashion and written in standard English?

Reviewer #1: No

5. Review Comments to the Author

Reviewer #1: 1. “Page 1”

“However, due to the particularity of the basement concrete structure, crack control still faces some challenges[10].”

It is suggested to explain what is the particularity of the basement concrete structure.

2. “Page 2”

“2 Literature Review”

It only mentions the research status of crack identification in this paper, lacking the research status of greenhouse effect analysis methods. It is suggested to replenish accordingly.

3. “Page 3”

“Compared with traditional image processing methods, the Mask-RCNN algorithm has higher accuracy and stability, which can effectively solve the challenges in crack identification and fill the research gap.”

There are no comparisons in the text with the results of traditional method studies, so please check it or make additions.

4. “Page 3”

“The mask-RCNN model consists of three main components: Backbone Network, Region Proposal Network (RPN), and Mask Branch. The structure composition is displayed in Figure 1:”

It is recommended that the backbone, Regional Proposed Network (RPN) and masked branches be labelled on Figure 1.

5. “Page 3”

“Figure 1 signifies that Backbone Network usually uses pre-trained CNNs, such as ResNet or VGGNet, to extract image features.”

It is recommended to state whether ResNet or VGGNet was used in this study.

6. “Page 3”

“The following steps are used for preprocessing:”

It is recommended to give a more specific treatment process and provide a comparison chart of the sample before and after treatment.

7. “Page 6”

“Figure 6”

The description of Figure 6 in this study is incomplete, and it is suggested that it be supplemented. It can’t be seen in Figure 6 that with the increase of temperature in different experiments, the maximum stress of the structure presents a decreasing trend, please check it.

8. “Page 7”

“Figure 9”

It is suggested that a graphical note be given in Figure 9 to indicate which experiment 1 to 5 are respectively.

9. “Page 7”

“Figure 10”

The description of Figure 10(a) is missing from this paper, please add it.

10. “Page 7”

“References”

Although more than 30 literatures are reviewed in this paper, most of them are from the past three years. There is a lack of review of earlier literatures, and literature research needs to be improved.

Finally, there are some grammatical errors and in some cases English is not used properly. The authors should pay attention to correct these mistakes throughout the manuscript.

6. PLOS authors have the option to publish the peer review history of their article (what does this mean?). If published, this will include your full peer review and any attached files.

Reviewer #1: No

---

## [Author Response · Author response to Decision Letter 0]

18 Sep 2023

Reviewer #1: 1. “Page 1”

“However, due to the particularity of the basement concrete structure, crack control still faces some challenges[10].”

It is suggested to explain what is the particularity of the basement concrete structure.

Reply: Thank you for your valuable feedback from the reviewers. There are indeed some challenges regarding the specificity of the concrete structure in the basement, and we strongly agree with this. Hence, I added an explanation in the introduction about the particularity of the concrete structure in the basement. Concrete structures in basements often face several special factors, such as the particularity of the underground environment, load and constraint conditions, and temperature effects.

2. “Page 2”

“2 Literature Review”

It only mentions the research status of crack identification in this paper, lacking the research status of greenhouse effect analysis methods. It is suggested to replenish accordingly.

Reply: Thank you for the reviewer's suggestion. We attach great importance to your feedback. In the article, We should supplement and clarify the research status of greenhouse effect analysis methods. In the revised manuscript, we have expanded the literature review section to include the latest research progress related to greenhouse effect analysis methods (in the past five years). This will help readers better understand the positioning and value of our research in the field of greenhouse effect analysis methods.

3. “Page 3”

“Compared with traditional image processing methods, the Mask-RCNN algorithm has higher accuracy and stability, which can effectively solve the challenges in crack identification and fill the research gap.”

There are no comparisons in the text with the results of traditional method studies, so please check it or make additions.

Reply: Thank you for the reviewer's suggestion. Your viewpoint is very valuable; indeed, there is no comparison with the research results of traditional methods in the article. We have added a comparison of experimental results with traditional methods in the revised manuscript (Figure 10) to support the superiority of our proposed Mask RCNN algorithm. By comparing the optimization performance with traditional image processing methods, including crack density, average crack length, average crack width, maximum stress concentration, and maximum deformation, it helps to more comprehensively demonstrate the advantages of the Mask RCNN algorithm in crack recognition.

4. “Page 3”

“The mask-RCNN model consists of three main components: Backbone Network, Region Proposal Network (RPN), and Mask Branch. The structure composition is displayed in Figure 1:”

It is recommended that the backbone, Regional Proposed Network (RPN) and masked branches be labelled on Figure 1.

Reply: Thank you for the reviewer's suggestion. We have added labels in Figure 1 of the article to clearly display the three main components of the Mask RCNN model: Backbone Network, Region Proposal Network (RPN), and Mask Branch, to help readers better understand the structural composition of the Mask-RCNN model. This can improve the readability and comprehension of the article.

5. “Page 3”

“Figure 1 signifies that Backbone Network usually uses pre-trained CNNs, such as ResNet or VGGNet, to extract image features.”

It is recommended to state whether ResNet or VGGNet was used in this study.

Reply: Thank you for the reviewer's suggestion. We have modified this sentence. Figure 1 shows that the backbone network uses pre-trained convolutional neural networks, where VGGNet is used to extract image features.

6. “Page 3”

“The following steps are used for preprocessing:”

It is recommended to give a more specific treatment process and provide a comparison chart of the sample before and after treatment.

Reply: Thank you for the reviewer's suggestion. In the revised draft, we provided more detailed preprocessing steps, including a specific description of the processing process, and added a comparison chart of the samples before and after processing (Figure 2) to more clearly demonstrate the effectiveness of the processing. These improvements help readers better understand our preprocessing process and the impact of processing on the samples. Thank you again for your suggestion. 

7. “Page 6”

“Figure 6”

The description of Figure 6 in this study is incomplete, and it is suggested that it be supplemented. It can’t be seen in Figure 6 that with the increase of temperature in different experiments, the maximum stress of the structure presents a decreasing trend, please check it.

Reply: Thank you for the reviewer's suggestion. We do realize that the description of Figure 6 (now Figure 7) is incomplete, so we have revised the form of Figure 6 in the revised draft to provide more detailed trend explanations and supplementary explanations to ensure that readers can better understand the content of the figure. Regarding the relationship between temperature and maximum stress, we have provided more context and explanation to demonstrate that in different experiments, the maximum stress of the structure does indeed show a decreasing trend with increasing temperature. This key observation has been better explained and presented in the revised draft.

8. “Page 7”

“Figure 9”

It is suggested that a graphical note be given in Figure 9 to indicate which experiment 1 to 5 are respectively.

Reply: Thank you for the reviewer's suggestion. I have modified the format of Figure 9 (now Figure 11) and added a legend to indicate Experiment 1 to Experiment 5. This legend will include identification and corresponding color coding for each experiment to ensure the readability and comprehensibility of the diagram.

9. “Page 7”

“Figure 10”

The description of Figure 10(a) is missing from this paper, please add it.

Reply: Thank you for the reviewer's correction. We have added a detailed description of Figure 10 (a) (now Figure 12) in the revised draft to ensure a clear explanation and interpretation of the figure. This will include data, trends, and information related to key findings in the study in the chart.

10. “Page 7”

“References”

Although more than 30 literatures are reviewed in this paper, most of them are from the past three years. There is a lack of review of earlier literatures, and literature research needs to be improved.

Reply: Thank you for the guidance of the reviewer. We are very grateful for your valuable feedback. We fully agree with your viewpoint on the shortcomings of the literature review. In the revised manuscript, we delved into and integrated earlier literature in the literature review section to provide a more comprehensive historical background and tracking of research development. This will help readers better understand the role and value of this study and our contributions to related fields.

Finally, there are some grammatical errors and in some cases English is not used properly. The authors should pay attention to correct these mistakes throughout the manuscript.

Reply: Thank you for the reviewer's review and correction. We recognize that there are some grammar errors and inappropriate English expressions in the article. In the revised manuscript, we have carefully reviewed each section of the article, paying special attention to grammar, spelling, and expression issues, and consulted professional language editors to improve the readability and professionalism of the article, striving to ensure that it meets the standards.

---

## [Decision Letter · Decision Letter 1]

20 Sep 2023

Crack Control Optimization of Basement Concrete Structures Using the Mask-RCNN and Temperature Effect Analysis

PONE-D-23-21940R1

Dear Dr. Fu,

We’re pleased to inform you that your manuscript has been judged scientifically suitable for publication and will be formally accepted for publication once it meets all outstanding technical requirements.

Kind regards,

Paul Awoyera

Academic Editor

PLOS ONE

Additional Editor Comments (optional):

Reviewers' comments:

Reviewer's Responses to Questions

**Comments to the Author**

1. If the authors have adequately addressed your comments raised in a previous round of review and you feel that this manuscript is now acceptable for publication, you may indicate that here to bypass the “Comments to the Author” section, enter your conflict of interest statement in the “Confidential to Editor” section, and submit your "Accept" recommendation.

Reviewer #1: All comments have been addressed

2. Is the manuscript technically sound, and do the data support the conclusions?

Reviewer #1: Yes

3. Has the statistical analysis been performed appropriately and rigorously? 

Reviewer #1: Yes

4. Have the authors made all data underlying the findings in their manuscript fully available?

Reviewer #1: Yes

5. Is the manuscript presented in an intelligible fashion and written in standard English?

Reviewer #1: Yes

6. Review Comments to the Author

Reviewer #1: The authors have adequately addressed the comments. Manuscript improved quality. Now, it is acceptable.

7. PLOS authors have the option to publish the peer review history of their article (what does this mean?). If published, this will include your full peer review and any attached files.

Reviewer #1: No

---

## [Editor Report · Acceptance letter]

26 Sep 2023

PONE-D-23-21940R1 

Crack Control Optimization of Basement Concrete Structures Using the Mask-RCNN and Temperature Effect Analysis 

Dear Dr. Fu:

I'm pleased to inform you that your manuscript has been deemed suitable for publication in PLOS ONE. Congratulations! Your manuscript is now with our production department. 

Kind regards, 

on behalf of

Dr. Paul Awoyera 

Academic Editor

PLOS ONE